# LIGHT-SEARCH: REDUCING RETRIEVAL COST IN RAG VIA CURRICULUM-BASED POLICY OPTIMIZATION

## ABSTRACT

Retrieval-Augmented Generation (RAG) is pivotal for modern Large Language Models. However, its practical deployment is often hindered by prohibitive inference costs, encompassing both latency and financial overhead from retrieval calls. Current reinforcement learning frameworks focus on improving search capability by solely maximizing answer accuracy, which inadvertently encourages excessive and costly search behavior. This overlooks the fundamental trade-off between task performance and computational efficiency. To address this, we introduce *Light-Search*, a systematic reinforcement learning framework that teaches models to balance answer quality with search cost. We find that naively penalizing search actions leads to unstable training and suboptimal policies. Therefore, *Light-Search* employs a *two-stage curriculum* that first builds robust search capabilities before introducing a cost-augmented reward function to cultivate efficiency. This learning process is underpinned by a stabilized policy optimization algorithm, ensuring the model can robustly learn a judicious policy on when to search. Experiments across diverse question-answering benchmarks show that *Light-Search* drastically reduces retrieval calls by up to 76.5% while maintaining performance competitive with state-of-the-art models. By enabling a controllable balance between effectiveness and efficiency, *Light-Search* provides a practical path toward building powerful yet economical RAG systems.

## 1 INTRODUCTION

Large Language Models (LLMs) have demonstrated remarkable capabilities in natural language understanding and generation (Achiam et al., 2023). However, their knowledge is static, confined to the data they were trained on, leading to factual inaccuracies ("hallucinations") and an inability to access real-time information. Retrieval-Augmented Generation (RAG) has emerged as a powerful paradigm to mitigate these limitations, dynamically retrieving relevant documents from external knowledge sources to ground the generation process (Fan et al., 2024; Lewis et al., 2020; Guu et al., 2020). This approach has achieved state-of-the-art performance on a wide array of knowledge-intensive tasks.

Despite its success, the practical deployment of RAG is severely constrained by a critical factor: the inference cost of the retrieval step. Each retrieval operation introduces significant latency and incurs direct financial costs when using commercial search APIs (Xu & Peng, 2025) as shown in Figure 1. In a real-world, high-throughput environment, these costs accumulate rapidly, rendering many sophisticated RAG systems economically and operationally infeasible. The key is not just to make RAG *effective*, but to make it *efficient*.

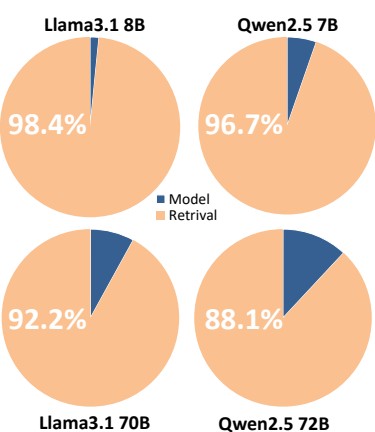

Figure 1: Breakdown of average inference cost per query for RAG systems. Retrieval costs (orange) dominate the cost compared to model inference (blue), highlighting the need for efficient retrieval strategies.

A promising direction for optimizing this process is to make the retrieval decision adaptive. Reinforcement Learning (RL) presents a natural framework for this, modeling the LLM as an agent

that learns a sequential policy on when to search for information versus when to generate an answer directly (Jiang et al., 2023; Asai et al., 2024). However, prevailing RL-based approaches are *myopic* in their formulation. They typically design reward functions that exclusively target the maximization of final answer accuracy. This inadvertently trains the agent to adopt a "search-at-all-costs" strategy, as more evidence often correlates with a higher chance of a correct answer. While this may improve task *capability*, it fundamentally fails to address the underlying efficiency problem.

In this work, we argue that an effective RAG agent must learn to explicitly navigate the trade-off between task performance and computational cost. To this end, we introduce *Light-Search*, a systematic RL framework designed to train cost-aware RAG agents. A naive solution might involve simply adding a fixed penalty for each search action to the reward function. However, our initial investigations reveal that this approach is highly unstable, often leading to training collapse or suboptimal "lazy" policies where the agent learns to avoid searching altogether.

To overcome this instability, *Light-Search* employs a carefully designed *two-stage curriculum* with *Two-Stage Advantage Shaping (TSAS)*. The first stage, a Capability-Building (Warm-up) phase, focuses on developing search competence. Here, the advantage function is shaped to be cost-agnostic, rewarding exploration and reasoning to build a robust understanding of how to effectively use the retrieval tool. The second stage, an Efficiency-Cultivating (Annealing) phase, transitions the objective to cost-awareness. It introduces a novel advantage function where a soft performance gate, conditioned on the group-relative quality of the generated answer, dynamically modulates the reward. For high-quality answers, the associated search behavior is rewarded, reinforcing effective information seeking. Conversely, for low-quality answers, the search is penalized, discouraging redundant retrieval. This entire curriculum is optimized using our *Light-Search Group-Relative Policy Optimization (LS-GRPO)* algorithm, which robustly teaches the agent a judicious policy on when to search.

We conduct extensive experiments on diverse question-answering benchmarks, from single-hop QA (Kwiatkowski et al., 2019; Joshi et al., 2017) to complex multi-hop reasoning (Yang et al., 2018; Ho et al., 2020). The results demonstrate that *Light-Search* achieves significant efficiency gains, reducing retrieval calls by up to 76.5%. Crucially, this is achieved with a negligible impact on task performance, maintaining accuracy competitive with state-of-the-art RAG models.

Our contributions are as follows:

- We identify and formalize a critical flaw in existing RL-based RAG frameworks: their myopic focus on accuracy leads to inefficient, costly search policies unsuited for practical deployment.

- We propose a systematic two-stage curriculum learning framework that decouples capability-building from efficiency-tuning, proving essential for stable training.

- We design a novel adaptive policy optimization mechanism that synergizes a performance-gated reward function with a stabilized group-based algorithm, enabling the model to robustly learn a cost-efficient policy.

- We provide comprehensive empirical validation across seven diverse QA benchmarks, demonstrating that *Light-Search* drastically reduces retrieval costs while maintaining competitive performance.

## 2 RELATED WORK

### 2.1 RETRIEVAL-AUGMENTED GENERATION

To mitigate the factual inaccuracies inherent in static Large Language Models (LLMs) and grant them access to real-time information, the RAG paradigm was introduced (Lewis et al., 2020; Guu et al., 2020). Early RAG systems employed a fixed "retrieve-then-read" pipeline, where documents were fetched once and then consumed by the generator (Izacard & Grave, 2021). However, it was soon recognized that not all queries require retrieval, and complex questions often necessitate an iterative, multi-step information-seeking process (Mallen et al., 2022). This insight spurred the development of adaptive retrieval methods.

Initial progress towards adaptive retrieval was made via sophisticated prompt engineering, such as Chain-of-Thought (Wei et al., 2022) and its derivatives, which manually structure the model's reasoning and search steps (Press et al., 2022; Yoran et al., 2023). To move beyond brittle, hand-crafted prompts, the field turned to learned approaches. Supervised fine-tuning (SFT) was used to train models on expert-annotated search trajectories. A prominent example is Self-RAG (Asai et al., 2024), which introduces special "reflection" tokens to teach a model to decide for itself whether to retrieve information and how to ground its generation. Other works have focused on optimizing the retriever for a given LLM (Shi et al., 2023) or even replacing retrieval with generation (Yu et al., 2022). While these methods advanced RAG capabilities, they are fundamentally constrained by their reliance on expensive and often limited demonstration data. In contrast, our framework utilizes reinforcement learning to learn a dynamic retrieval policy, obviating the need for static rules or expert-annotated trajectories.

## 2.2 LEARNING TO SEARCH WITH REINFORCEMENT LEARNING

RL provides a powerful framework for training LLMs as autonomous agents capable of learning complex behaviors through trial and error (Hou et al., 2025). Recently, a surge of research has applied RL to teach LLMs how to use tools, particularly web search, to solve complex problems. For instance, Active RAG (Jiang et al., 2023) explored using RL to learn a policy that decides whether to perform another retrieval during the generation process. After the release of OpenAI Deep Research (OpenAI, 2025), a series of works including Search-R1 (Jin et al., 2025), R1-Searcher (Song et al., 2025), WebThinker (Li et al., 2025), and DeepResearcher (Zheng et al., 2025) have demonstrated that RL can enable agents to discover sophisticated, multi-hop search and reasoning strategies that outperform SFT-based methods on knowledge-intensive tasks (Xu & Peng, 2025).

However, a critical and unifying limitation of these capability-focused RL methods is that they are designed almost exclusively to maximize final answer accuracy. In contrast, our work is the first to systematically incorporate cost into the RL objective, training an agent to explicitly balance task performance with search efficiency.

## 2.3 EFFICIENT LARGE LANGUAGE MODELS

Given the substantial computational demands of LLMs, a vibrant research area is dedicated to improving their inference efficiency (Wan et al., 2023). These efforts primarily focus on reducing the model's intrinsic computational cost. Key techniques include model compression, such as quantization (reducing numerical precision), pruning (removing redundant weights), and knowledge distillation (training a smaller model to mimic a larger one) (Zhu et al., 2024; Xu et al., 2024). Another major direction is the development of efficient architectures, most notably through novel attention mechanisms that approximate the standard quadratic-complexity self-attention with more scalable, linear-time alternatives (Sun et al., 2025b). The common goal of these methods is to lower the latency, memory, and energy consumption of a single forward pass of the model.

Our work is *orthogonal* to these model-centric optimizations. While they reduce the computational cost of a single forward pass, we focus on improving the *strategic efficiency* of an agent's policy to minimize the number of costly external actions (e.g., API calls).

## 3 METHODOLOGY: THE LIGHT-SEARCH FRAMEWORK

To train a cost-aware RAG agent, we propose *Light-Search*, a comprehensive training framework built upon a *Two-Stage Curriculum Learning* strategy. This strategy is operationalized through two primary components: 1) *Two-Stage Advantage Shaping (TSAS)*, which defines the precise reward and advantage structure for each curriculum stage, and 2) a policy optimization algorithm *Light-Search Group-Relative Policy Optimization (LS-GRPO)*, which we stabilize for the curriculum setting.

### 3.1 PROBLEM FORMULATION

We formulate the task of controlling a RAG agent as a finite-horizon Markov Decision Process (MDP), defined by the following components:

- **State** ($s_t$)**:** The state at timestep $t$ is a tuple $s_t = (\mathbf{q}, h_{t-1})$, where $\mathbf{q}$ is the initial user query and $h_{t-1}$ is the history of past actions and their corresponding observations.

- **Action** ($a_t$)**:** At each step, the agent chooses an action from a discrete set $\mathcal{A} = \{\text{SEARCH(QUERY)}, \text{GENERATE(ANSWER)}\}$.

- **Policy** ($\pi_\theta$)**:** The agent's policy $\pi_\theta(a_t|s_t)$ is represented by an LLM with parameters $\theta$.

- **Trajectory** ($\tau$)**:** A trajectory is a full sequence of states and actions, $\tau = (s_0, a_0, \ldots, s_T, a_T)$, generated by the policy.

- **Reward and Advantage:** At the end of a trajectory $\tau$, the agent receives a scalar reward, $R(\tau)$, which evaluates the quality of the final output. The fundamental goal is to learn a policy $\pi_\theta$ that maximizes the expected total reward, $\mathbb{E}_{\tau \sim \pi_\theta}[R(\tau)]$. However, policy gradient methods suffer from high variance when using raw rewards. To stabilize training, modern algorithms learn from the *advantage*, $A(\tau)$, which measures how much better a trajectory's reward is than a baseline (e.g., the average reward). The policy is updated to favor actions leading to positive advantage. The core of our contribution lies in how we shape this advantage $A(\tau)$ through a curriculum.

A central challenge in training cost-aware agents is the cold start policy collapse problem. A naive approach might use a single, composite objective from the outset, aiming to maximize a reward like $R(\tau) - \lambda \cdot C(\tau)$, where $C(\tau)$ is the trajectory cost. However, for a weakly initialized agent that has yet to acquire effective search skills, simultaneously maximizing $R(\tau)$ and minimizing $C(\tau)$ is noisy and challenging; reward hacking often emerges, the gradient is soon dominated by the cost penalty, and the agent collapses into a "lazy" policy that trivially maximizes its objective. This leads to a suboptimal local minimum where the agent never acquires task-solving skills. To circumvent this, we propose a *two-stage curriculum* that decouples capability acquisition from efficiency cultivation.

### 3.1.1 STAGE 1: CAPABILITY-BUILDING (WARM-UP)

The exclusive objective of this stage is to develop a competent agent. Conceptually, we aim to find a "capable policy," $\pi_{\text{capable}}$, that maximizes a composite reward encouraging both task success and exploration. This can be formally described as:

$$\pi_{\text{capable}} = \arg\max_{\pi_\theta} \mathbb{E}_{\tau \sim \pi_\theta}[R_1(\tau)] \tag{1}$$

where $R_1(\tau) = R(\tau) + R_{\text{int}}(\tau)$. Here, $R(\tau)$ is the external task reward, and $R_{\text{int}}(\tau)$ is an intrinsic reward function that encourages exploratory behaviors causally linked to success, such as using tools and generating detailed reasoning. It is a function of the search count $S(\tau)$ and reasoning length $T_{\text{len}}(\tau)$.

### 3.1.2 STAGE 2: EFFICIENCY-CULTIVATING (ANNEALING)

This stage commences after the agent has developed a baseline competence. The objective shifts to finding an "efficient policy," $\pi_{\text{efficient}}$, that refines the strategies learned in Stage 1. This policy should maximize task reward while penalizing cost *only when it is not justified by performance*. We formalize this as:

$$\pi_{\text{efficient}} = \arg\max_{\pi_\theta} \mathbb{E}_{\tau \sim \pi_\theta}[R_2(\tau)] \tag{2}$$

where $R_2(\tau) = R(\tau) - g(R(\tau)) \cdot C(\tau)$. In this formulation, $C(\tau)$ is the trajectory cost (a function of $S(\tau)$ and $T_{\text{len}}(\tau)$), and $g(R(\tau))$ is a crucial *performance gating function*. This gate is designed to be near-zero for failing trajectories (where $R(\tau)$ is low), thus ignoring their cost, but becomes positive for successful trajectories, creating pressure to find more cost-effective solutions.

The transition between these two stages is triggered after a fixed number of training iterations, $M_1$. This curriculum design directly motivates our specific advantage shaping mechanism.

### 3.2 TWO-STAGE ADVANTAGE SHAPING (TSAS)

TSAS implements the conceptual goals of our curriculum by defining two different advantage functions, $\hat{A}_i$, one for each stage. Let's denote the $i$-th trajectory in a batch of size $G$ as $(\mathbf{q}, \mathbf{o}_i)$.

**Stage 1: Capability-Building Advantage.** In the first stage, the advantage function $\hat{A}_{i,t}$ for trajectory $i$ at timestep $t$ is defined as:

$$\hat{A}_{i,t} \equiv R(\mathbf{q}, \mathbf{o}_i) - \text{mean}\{R(\mathbf{q}, \mathbf{o}_j)\}_{j=1}^{G} + \log\big(S(\mathbf{q}, \mathbf{o}_i) + 1\big) + \alpha \log\big(T_{\text{len}}(\mathbf{q}, \mathbf{o}_i)\big) \tag{3}$$

This function combines the group-normalized task reward with intrinsic rewards for search ($S$) and reasoning ($T_{\text{len}}$), directly reflecting the goal of the capability-building stage. Specifically, this advantage function combines the group-relative reward $R(\mathbf{q}, \mathbf{o}_i)$ with auxiliary rewards: $S(\mathbf{q}, \mathbf{o}_i)$ counts the number of search operations performed, and $T_{\text{len}}(\mathbf{q}, \mathbf{o}_i)$ measures the total length of reasoning traces. The hyperparameter $\alpha$ controls the relative importance of reasoning depth, while $\epsilon$ defines the clipping range for stable optimization.

**Stage 2: Efficiency-Cultivating (Annealing) Advantage.** In the second stage, the advantage function is reformulated to implement the performance-gated efficiency objective:

$$\hat{A}_i \equiv A_i^+ + \sigma_i \cdot S_i^+ + \sigma_i \cdot T_i^+ \tag{4}$$

The components are defined as follows:

- $A_i^+$: The group-normalized task reward, defined as $R(\mathbf{q}, \mathbf{o}_i) - \text{mean}\, R(\mathbf{q}, \mathbf{o}_j)j = 1^G$.
- $S_i^+$: A normalized reward for searching, calculated as the logarithm of the search count minus the batch's standard deviation of search counts: $\log(S(\mathbf{q}, \mathbf{o}_i) + 1) - \text{std}\,\cdot$.
- $T_i^+$: A reward for reasoning length, given by the logarithm of the trajectory's token length: $\log(T_{\text{len}}(\mathbf{q}, \mathbf{o}_i) + 1)$.
- $\sigma_i$: The critical *soft performance gate*, $\text{sigmoid}(A_i^+) - 0.5$. When a trajectory is successful ($A_i^+ > 0$), $\sigma_i$ is positive, preserving the intrinsic rewards from $S^+ i$ and $T_i^+$. When it fails ($A_i^+ < 0$), $\sigma_i$ becomes negative, effectively penalizing costly exploration that did not lead to success.

### 3.3 A Two-Stage Curriculum for Cost-Awareness

The key insight of this formulation is its dynamic, self-annealing nature. Early in training, when score distributions are uniform, the mechanism promotes exploration by rewarding costly actions in successful trajectories and penalizing them in failures. As the policy matures and score distributions become skewed and the penalties for the majority of low-scoring trajectories become the dominant signal. This naturally transitions the training objective from balancing performance and cost toward prioritizing efficiency, guiding the model to an optimal and stable search behavior that evolves with its capabilities.

Figure 2: Evolution of the group-relative score distribution during annealing: as training progresses (red to blue), the shape shifts from slightly left-skewed to right-skewed while simultaneously sharpening.

### 3.4 LS-GRPO

To optimize the policy $\pi_\theta$ using the advantages defined by TSAS, we employ a policy gradient algorithm based on GRPO (), which we refer to as *Light-Search* Group-Relative Policy Optimization (LS-GRPO). The objective function for Stage 1 is:

$$\mathcal{J}_{\text{LS-GRPO}}(\pi_\theta) = \mathbb{E}_{\mathbf{q} \sim p_\mathcal{Q},\, \{\mathbf{o}_i\}_{i=1}^{G} \sim \pi_{\theta_{\text{old}}}(\cdot|\mathbf{q})} \frac{1}{G} \sum_{i=1}^{G} \sum_{t=1}^{|\mathbf{o}_i|} \Big\{ \min\Big[ \rho_{i,t} \hat{A}_{i,t},\ \text{clip}\big(\rho_{i,t}, 1-\epsilon, 1+\epsilon\big) \hat{A}_{i,t} \Big] \Big\}, \tag{5}$$

where $\mathbf{q} \sim p_\mathcal{Q}$ represents queries sampled from the task distribution, $\{\mathbf{o}_i\}_{i=1}^{G}$ denotes a group of $G$ responses generated by the reference policy $\pi_{\theta_{\text{old}}}$, $\hat{A}_{i,t}$ is the advantage from Eq. 3. The objective for

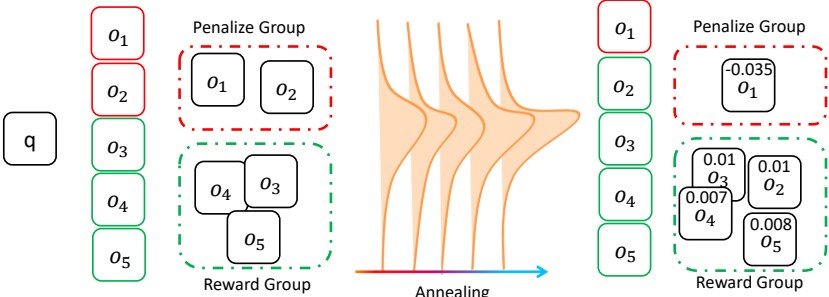

Figure 3: Evolution of the Group Relative Score distribution for $A_i^+$. $q$ denotes the query; $o_1 \sim o_5$ are the model-sampled group responses.

| Method | Single-Hop QA | | | | | | | | | Averages | |
| | NQ | | | TriviaQA | | | PopQA | | | Avg Acc | Avg ST |
| | Acc | ST | SD | Acc | ST | SD | Acc | ST | SD | | |
|---|---|---|---|---|---|---|---|---|---|---|---|
| *Qwen-2.5-7B* | | | | | | | | | | | |
| Direct Answer-base | 12.40 | - | - | 21.80 | - | - | 7.20 | - | - | 13.80 | - |
| Direct Answer-instruct | 11.60 | - | - | 35.60 | - | - | 1.20 | - | - | 16.13 | - |
| CoT-base | 21.40 | - | - | 34.60 | - | - | 13.00 | - | - | 23.00 | - |
| CoT-instruct | 27.00 | - | - | 45.20 | - | - | 15.00 | - | - | 29.07 | - |
| RAG-base | 20.60 | 0.68 | 0.81 | 31.60 | 0.67 | 0.79 | 22.20 | 0.52 | 0.67 | 24.80 | 0.62 |
| RAG-instruct | 20.20 | **0.07** | **0.25** | 28.20 | **0.11** | **0.32** | 22.20 | **0.03** | **0.18** | 23.53 | **0.07** |
| ZeroSearch | 41.60 | 0.89 | 0.61 | 57.80 | 0.92 | 0.62 | 50.40 | 0.84 | 0.48 | 49.93 | 0.88 |
| *Light-Search* | **44.40** | 0.94 | 0.41 | **64.00** | 0.83 | 0.52 | **59.20** | 0.82 | 0.42 | **55.87** | 0.86 |
| *Qwen-2.5-3B* | | | | | | | | | | | |
| Direct Answer-base | 7.00 | - | - | 14.40 | - | - | 4.00 | - | - | 8.47 | - |
| Direct Answer-instruct | 16.20 | - | - | 26.60 | - | - | 14.40 | - | - | 19.07 | - |
| CoT-base | 9.00 | - | - | 13.60 | - | - | 6.00 | - | - | 9.53 | - |
| CoT-instruct | 19.40 | - | - | 35.60 | - | - | 8.20 | - | - | 21.07 | - |
| RAG-base | 10.40 | 0.50 | 0.63 | 16.20 | 0.57 | 0.69 | 11.40 | 0.59 | 0.67 | 12.67 | 0.55 |
| RAG-instruct | 16.20 | 0.18 | **0.41** | 28.20 | 0.24 | 0.46 | 25.60 | 0.36 | 0.49 | 23.33 | 0.26 |
| ZeroSearch | 44.60 | 0.51 | 0.50 | 64.60 | 0.21 | 0.41 | 64.60 | 0.30 | 0.46 | 57.93 | 0.34 |
| *Light-Search* | **48.00** | **0.23** | 0.43 | **65.80** | **0.02** | **0.15** | **66.20** | **0.00** | **0.00** | **60.00** | **0.08** |
| *LLaMA-3.2-3B* | | | | | | | | | | | |
| Direct Answer | 27.40 | - | - | 51.40 | - | - | 23.80 | - | - | 34.20 | - |
| CoT | 26.20 | - | - | 44.40 | - | - | 2.80 | - | - | 24.47 | - |
| RAG | 28.80 | **0.78** | 0.74 | 45.60 | **0.68** | 0.68 | 35.80 | **0.62** | 0.54 | 36.73 | **0.69** |
| ZeroSearch | 38.20 | 0.89 | 0.32 | 55.60 | 0.85 | 0.35 | 57.20 | 0.90 | 0.30 | 50.33 | 0.88 |
| *Light-Search* | **47.80** | 1.04 | **0.21** | **67.00** | 1.02 | **0.19** | **73.20** | 1.01 | **0.11** | **62.67** | 1.02 |

Table 1: Main results for Single-Hop QA tasks using different LLMs as the backbone. The best performance is set in bold. Acc: Accuracy (%), ST: Search Times, SD: Search Standard Deviation.

Stage 2 uses the same structure, but substitutes $\hat{A}_{i,t}$ with the annealing advantage $\hat{A}_i$ from Eq. 4. and $\rho_{i,t}$ is the importance sampling ratio between the current and reference policies.

$$\rho_{i,t} = \frac{\pi_\theta(o_{i,t}|\mathbf{q}, \mathbf{o}_{i,<t})}{\pi_{\theta_{old}}(o_{i,t}|\mathbf{q}, \mathbf{o}_{i,<t})} \tag{6}$$

The clipping term, controlled by $\epsilon$, ensures stable training. This combination of a clear curriculum, precisely shaped advantages, and a stable optimization algorithm is key to our framework.

# 4  EXPERIMENT

## 4.1  EXPERIMENT SETUP

Our framework is built upon the *verl* Sheng et al. (2024), which is optimized for distributed reinforcement learning with large models. All experiments are conducted on a single node equipped with 8

| Method | Multi-Hop QA | | | | | | | | | | | | Averages | |
|---|---|---|---|---|---|---|---|---|---|---|---|---|---|---|
| | HotpotQA | | | 2Wiki | | | Musique | | | Bamboogle | | | Avg Acc | Avg ST |
| | Acc | ST | SD | Acc | ST | SD | Acc | ST | SD | Acc | ST | SD | | |
| *Qwen-2.5-7B* | | | | | | | | | | | | | | |
| Direct Answer-base | 11.40 | - | - | 14.20 | - | - | 2.60 | - | - | 6.94 | - | - | 8.79 | - |
| Direct Answer-instruct | 16.40 | - | - | 22.20 | - | - | 4.80 | - | - | 14.40 | - | - | 14.45 | - |
| CoT-base | 14.80 | - | - | 21.40 | - | - | 6.80 | - | - | 13.89 | - | - | 14.22 | - |
| CoT-instruct | 21.00 | - | - | 24.80 | - | - | 8.00 | - | - | 26.39 | - | - | 20.05 | - |
| RAG-base | 23.00 | 0.76 | 0.88 | 19.40 | 0.92 | 0.96 | 8.00 | 0.85 | 0.91 | 18.06 | 0.81 | 0.95 | 17.12 | 0.84 |
| RAG-instruct | 17.40 | **0.28** | **0.51** | 19.20 | **0.63** | 0.80 | 7.40 | **0.26** | **0.55** | 26.39 | **0.07** | **0.25** | 17.60 | **0.31** |
| ZeroSearch | 32.80 | 1.20 | 0.77 | 32.20 | 1.49 | 0.93 | 19.00 | 1.30 | 0.80 | 44.00 | 1.18 | 0.74 | 32.00 | 1.29 |
| *Light-Search* | **34.00** | 1.10 | 0.60 | **41.00** | 1.39 | **0.69** | **21.00** | 1.25 | 0.66 | 36.00 | 1.27 | 0.69 | **33.00** | 1.25 |
| *Qwen-2.5-3B* | | | | | | | | | | | | | | |
| Direct Answer-base | 7.40 | - | - | 8.40 | - | - | 0.80 | - | - | 4.17 | - | - | 5.19 | - |
| Direct Answer-instruct | 17.00 | - | - | 19.00 | - | - | 4.20 | - | - | 2.78 | - | - | 10.75 | - |
| CoT-base | 6.40 | - | - | 9.40 | - | - | 1.00 | - | - | 2.78 | - | - | 4.90 | - |
| CoT-instruct | 15.60 | - | - | 21.00 | - | - | 4.80 | - | - | 19.44 | - | - | 15.21 | - |
| RAG-base | 7.80 | 0.68 | 0.74 | 9.80 | 0.68 | 0.78 | 1.20 | 0.65 | 0.78 | 5.56 | 0.64 | 0.63 | 6.09 | 0.66 |
| RAG-instruct | 16.60 | 0.50 | 0.65 | 21.80 | 0.68 | 0.73 | 8.00 | 0.54 | 0.69 | 11.11 | 0.44 | 0.62 | 14.38 | 0.54 |
| ZeroSearch | **37.80** | 0.26 | 0.45 | 34.20 | 0.93 | 0.36 | 18.20 | 0.89 | **0.32** | 22.22 | **0.00** | **0.00** | **28.11** | 0.52 |
| *Light-Search* | 37.40 | 0.51 | 0.50 | **37.20** | **0.00** | **0.00** | 18.40 | 0.26 | 0.44 | 16.00 | **0.00** | **0.00** | 27.25 | **0.19** |
| *LLaMA-3.2-3B* | | | | | | | | | | | | | | |
| Direct Answer | 19.60 | - | - | 21.60 | - | - | 4.00 | - | - | 6.94 | - | - | 13.04 | - |
| CoT | 16.00 | - | - | 10.20 | - | - | 5.80 | - | - | 21.60 | - | - | 13.40 | - |
| RAG | 18.60 | 0.95 | 0.82 | 14.80 | 1.18 | 0.87 | 7.20 | 0.95 | 0.86 | 19.44 | **0.81** | 0.70 | 15.01 | 0.97 |
| ZeroSearch | 22.20 | **0.87** | 0.33 | 23.00 | **0.87** | 0.34 | 9.20 | **0.91** | 0.29 | 18.06 | 0.86 | 0.35 | 18.12 | **0.88** |
| *Light-Search* | 33.20 | 1.07 | **0.25** | **34.80** | 1.04 | **0.21** | **15.20** | 1.11 | 0.31 | 21.60 | 1.02 | **0.15** | **26.20** | 1.06 |

Table 2: Main results for Multi-Hop QA tasks using different LLMs as the backbone. The best performance is set in bold. Acc: Accuracy (%), ST: Search Times, SD: Search Standard Deviation.

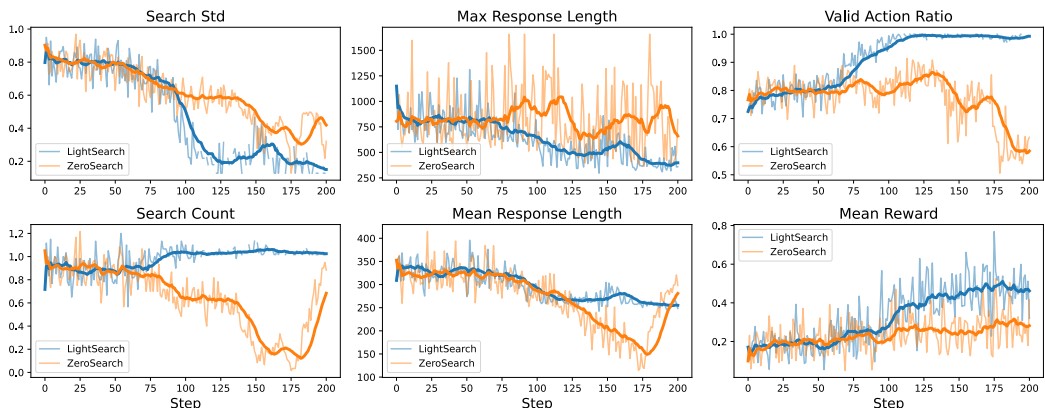

Figure 4: RL fine-tuning dynamics of *Light-Search* and *ZeroSearch*. Solid lines are moving averages over 15 steps.

NVIDIA A100 40GB GPUs. For complete hyperparameter configurations and other implementation details, please refer to appendix B.

**Datasets and Evaluation.** Following the setting of ZeroSearch Sun et al. (2025a), we use the ZeroSearch dataset for training our models. For evaluation, we assess performance on a diverse suite of seven question-answering benchmarks, which are divided into two categories: single-hop and multi-hop. A total of seven datasets are used (Kwiatkowski et al., 2019; Joshi et al., 2017; Mallen et al., 2022; Yang et al., 2018; Ho et al., 2020; Trivedi et al., 2022; Press et al., 2022); the details are reported in the appendix C.1. This allows us to measure both in-domain and out-of-domain generalization. Across all benchmarks, the F1 score is used as the performance reward for each answer during training. At the evaluation stage, Exact Match (EM) is used as the primary evaluation metric. To assess search cost and the stability of search behavior, we introduce two additional metrics: the number and standard deviation of searches (ST and SD).

**Models and Baselines.** To evaluate the robustness and generalizability of our findings, our experiments utilize several backbone language models from two distinct model families and at varying

| Method | TriviaQA | | | HotpotQA | | | Musique | | | Average | |
|---|---|---|---|---|---|---|---|---|---|---|---|
| | Acc | ST | SD | Acc | ST | SD | Acc | ST | SD | Avg. Acc | Avg. ST |
| *Light-Search* (Full) | **67.00** | **1.02** | **0.19** | **33.20** | **1.07** | **0.25** | **15.20** | **1.11** | **0.31** | **38.47** | **1.07** |
| w/o Stage 2 (Annealing) | 52.20 | 1.38 | 0.74 | 23.80 | 1.48 | 0.83 | 8.40 | 1.59 | 0.83 | 28.13 | 1.48 |
| w/o Stage 1 (Warm-up) | 61.80 | 1.22 | 0.51 | 30.60 | 1.28 | 0.53 | 12.60 | 1.37 | 0.59 | 35.00 | 1.29 |

Table 3: Ablation study on *LS-GRPO* using LLaMA-3.2-3B-Instruct. We evaluate the impact of removing key training stages: the annealing stage (Stage 2) and the warm-up stage (Stage 1). The results demonstrate that both stages are crucial for achieving optimal performance and search efficiency. Acc: Accuracy (%), ST: Average Search Times per query, SD: Search Standard Deviation.

scales. Specifically, we employ models from the *Qwen2.5 family* Qwen et al. (2025) at both *3B* and *7B* variants, and the *LLaMA-3.2-3B* model Dubey et al. (2024) from the LLaMA family. Our evaluation includes a comprehensive set of baselines: Direct Answer, Chain-of-Thought (CoT) Wei et al. (2022), standard Retrieval-Augmented Generation (RAG) Lewis et al. (2020), and ZeroSearch Sun et al. (2025a). For prompt-based baselines (Direct Answer and CoT), we utilize Instruct models, as Base models often struggle to follow specific task instructions. For the reinforcement learning-based methods (ZeroSearch and our own), we evaluate with the Base model for qwen and the Instruct model for llama to assess the generalizability of the approach across different model types.

## 4.2 RESULTS

### 4.2.1 OVERALL PERFORMANCE

The results in Table 1 and Table 2 show that *Light-Search* establishes a more favorable performance-efficiency frontier. Specifically, it shows superior performance-cost trade-off, its generalizability, and its enhanced policy stability.

***Light-Search* Establishing a Superior Performance-Efficiency Frontier.** *Light-Search* establishes a superior trade-off between performance and efficiency across both single- and multi-hop tasks. This is demonstrated on Qwen models, where *Light-Search* often improves accuracy while simultaneously reducing search cost; e.g., with Qwen-2.5-7B on single-hop tasks, it achieves higher accuracy (55.87% vs. 49.93%) with fewer searches (0.86 vs. 0.88). The results with Llama-3.2-3B highlight a more nuanced policy, where a marginal increase in search cost is traded for substantial accuracy gains (+12.34 points on single-hop, +8.08 on multi-hop). This indicates that our curriculum fosters a policy that optimizes for the marginal utility of each search, making strategic investments for disproportionate performance returns rather than defaulting to a naive cost-minimization strategy.

***Light-Search* Generalizability Across Diverse Models and Task Complexities.** The advantages of *Light-Search* generalize across model architectures and scales. With Qwen-2.5-7B on single-hop tasks, *Light-Search* achieves *55.87%* accuracy with *0.86* searches, outperforming ZeroSearch, which scores *49.93%* with a slightly higher cost of *0.88*. This demonstrates an instance of achieving higher accuracy with lower computational overhead. The trend holds for the Qwen-2.5-3B model, where *Light-Search* maintains a performance lead on single-hop tasks and is competitive on multi-hop tasks while reducing search frequency by over *60%* (0.19 vs. 0.52 Avg ST). This consistent behavior across different models and task complexities validates the robustness of our training framework.

***Light-Search* Enhanced Policy Stability and Operational Reliability.** Beyond aggregate efficiency, *Light-Search* induces a more stable and reliable policy, which manifests in two ways. First, it exhibits lower variance in its search behavior. As shown by the "SD" metric, *Light-Search* consistently reduces the search standard deviation; for Llama-3.2-3B, the SD is reduced to *0.17* from ZeroSearch's *0.32*. This indicates a more predictable agent that applies a consistent strategy to similar problems.

Meanwhile, *Light-Search* demonstrates superior operational reliability by mitigating the *format collapse* issue observed in the baseline, where the agent's outputs progressively degrade and fail to adhere to the required action format. Our *format reward design* directly addresses this by explicitly rewarding correctly formatted actions. As illustrated in Figure 4, this design leads to a more stable, valid action ratio and smaller fluctuations in response length throughout training.

| Method | Single-Hop QA | | | | | | | | | Averages | |
|---|---|---|---|---|---|---|---|---|---|---|---|
| | NQ | | | TriviaQA | | | PopQA | | | Avg Acc | Avg ST |
| | Acc | ST | SD | Acc | ST | SD | Acc | ST | SD | | |
| *LLaMA-3.2-3B-Instruct* | | | | | | | | | | | |
| ZeroSearch | 38.20 | **0.89** | **0.32** | 55.60 | **0.85** | 0.35 | 57.20 | **0.90** | **0.30** | 50.33 | **0.88** |
| *ZeroSearch$^+$* | **47.20** | 1.07 | 0.33 | **63.80** | 1.04 | **0.29** | **72.40** | 1.00 | 0.17 | **61.13** | 1.04 |

Table 4: Supplemental study (Single-Hop QA) comparing our method with the original ZeroSearch. The best performance is set in bold. Acc: Accuracy (%), ST: Search Times, SD: Search Standard Deviation.

| Method | Multi-Hop QA | | | | | | | | | | | | Averages | |
|---|---|---|---|---|---|---|---|---|---|---|---|---|---|---|
| | HotpotQA | | | 2Wiki | | | Musique | | | Bamboogle | | | Avg Acc | Avg ST |
| | Acc | ST | SD | Acc | ST | SD | Acc | ST | SD | Acc | ST | SD | | |
| *LLaMA-3.2-3B-Instruct* | | | | | | | | | | | | | | |
| ZeroSearch | 22.20 | **0.87** | **0.33** | 23.00 | **0.87** | **0.34** | 9.20 | **0.91** | **0.29** | 18.10 | **0.86** | 0.35 | 18.13 | **0.88** |
| *ZeroSearch$^+$* | **29.40** | 1.11 | 0.41 | **23.80** | 1.19 | 0.50 | **14.80** | 1.10 | 0.38 | **26.40** | 1.06 | **0.26** | **23.60** | 1.12 |

Table 5: Supplemental study (Multi-Hop QA) comparing our method with the original ZeroSearch. The best performance is set in bold. Acc: Accuracy (%), ST: Search Times, SD: Search Standard Deviation.

### 4.2.2 ABLATION STUDY

To investigate the necessity of our two-stage curriculum, we conducted an ablation study with the results presented in Table 3. The experiments confirm that both stages are indispensable for achieving optimal results. Removing the final *annealing stage* cripples accuracy (e.g., from 35.87 to 28.13) and causes erratic, excessive searches. This occurs because the agent learns *how* to search but is never taught *when* to do so efficiently, as it is never exposed to a cost-aware objective. Conversely, omitting the initial *warm-up stage* also degrades performance by increasing search cost and variance. The premature cost penalty stifles exploration, preventing the agent from developing a robust base policy for subsequent optimization. Ultimately, the results demonstrate that the two stages are complementary: the warm-up is essential for building a capable foundation, while the annealing stage is critical for refining it into a cost-efficient policy.

### 4.2.3 COMPLEMENTARY STUDY

To investigate the generalizability and modularity of our two-stage curriculum, we conducted a complementary study. In this experiment, we integrated our Stage 1 (Warm-up) into the existing $ZeroSearch$ framework, denoted as $ZeroSearch^+$. The results, presented in Table 4 and Table 5, show a significant performance improvement. $ZeroSearch^+$ consistently outperforms the original $ZeroSearch$ across all tested single-hop and multi-hop datasets, with the average accuracy increasing from 50.33 to 61.13 on single-hop tasks and from 18.13 to 23.60 on multi-hop tasks. This demonstrates that our warm-up strategy is not only effective within our own framework but can also serve as a transferable module to enhance other RL-based methods. Yet, lacking an explicit annealing phase, $ZeroSearch^+$ remains sub-optimal in search budget and stability.

## 5 CONCLUSION

In this work, we addressed the critical challenge of inference cost in RAG systems, where existing reinforcement learning methods inadvertently promote inefficient, accuracy-at-all-costs search policies. We introduced *Light-Search*, a systematic framework that trains cost-aware agents by explicitly balancing task performance with search efficiency. Extensive experiments demonstrate that *Light-Search* drastically reduces retrieval calls across a diverse suite of question-answering benchmarks, all while maintaining competitive task accuracy. By successfully navigating the trade-off between effectiveness and efficiency, our work provides a practical and principled path toward developing powerful, yet economically viable, LLM agents for real-world deployment.

## ETHICS STATEMENT

Our work presents a technical framework, *Light-Search*, aimed at improving the efficiency of Retrieval-Augmented Generation (RAG) by reducing retrieval costs. The research is focused on the algorithmic optimization of a model's search policy. All experiments were conducted using publicly available language models and standard academic benchmarks, with no use of private or sensitive data. Our method optimizes the behavior of existing models and does not introduce new ethical concerns beyond those inherent to the base language models themselves.

## REPRODUCIBILITY STATEMENT

To ensure the reproducibility of our results, we will release the complete source code for our *Light-Search* framework, including scripts for training and evaluation, as well as all final model checkpoints. Our implementation is based on the `verl` library and follows standard experimental setups. We have provided comprehensive implementation details in Appendix B, including all hyperparameters, model configurations, datasets, and the hardware environment. The main experiments reported in this paper were conducted with a fixed random seed to facilitate direct replication of our findings.

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

## A   DECLARATION OF LLM USAGE

Throughout the preparation of this manuscript, large language models served only as linguistic aids. They were invoked solely to (1) enhance sentence clarity and fluency, (2) correct grammar and adjust style for better readability, and (3) propose alternative wordings that preserved the intended technical meaning. LLMs played no role in study design, data gathering, algorithm creation, experimental execution, or outcome interpretation. All methodological insights, implementation choices, and scientific conclusions were developed independently by the authors, who affirm that no new ideas, data, or claims were generated by these tools and that the intellectual substance of the work remains exclusively their own.

## B   IMPLEMENTATION DETAILS

### B.1   TRAINING INFRASTRUCTURE AND FRAMEWORK

We implement our Light-Search framework using the verl training infrastructure Sheng et al. (2024), which provides efficient distributed training capabilities for reinforcement learning with large language models. All experiments are conducted on a single node with 8 NVIDIA A100-SXM4-40GB GPUs interconnected via NVLink. The compute node is equipped with dual AMD EPYC 7742 64-Core Processors (256 CPU cores in total) and 512 MiB L3 cache, ensuring sufficient computational resources for both model training and search simulation.

### B.2   MODEL CONFIGURATION

We conduct experiments with multiple base models to validate the generalizability of our approach:

- **Primary Models**: Qwen2.5-3B (Base/Instruct) Yang et al. (2024), Qwen2.5-7B (Base/Instruct), and Llama-3.2-3B-Instruct Dubey et al. (2024)
- **Context Length**: Maximum prompt length of 4,096 tokens and maximum response length of 512 tokens
- **Generation Settings**: During rollout, we employ $n = 5$ parallel agents with temperature $T = 1.0$ for diverse response generation

### B.3   TWO-STAGE CURRICULUM TRAINING

#### B.3.1   STAGE 1: LEARNING TO SEARCH (WARM-UP)

In the first stage, we focus on developing the model's search and reasoning capabilities:

- **Training Steps**: 150 steps.
- **Reward Configuration**: Set $\alpha = 0.01$ for search and thinking length rewards (Equation 5)

#### B.3.2   STAGE 2: LEARNING WHEN NOT TO SEARCH (ANNEALING)

The annealing stage refines the model's selective search behavior:

- **Training Steps**: 52 steps for efficiency optimization
- **Dynamic Rewards**: Sigmoid activation with performance-based adjustment (Equation 4)

### B.4   OPTIMIZATION HYPERPARAMETERS

We employ the following optimization settings across both training stages:

- **Learning Rate**: $1 \times 10^{-6}$ with cosine decay schedule
- **Warm-up**: 95% of total steps for learning rate warm-up
- **Batch Sizes**: Training batch size of 12, validation batch size of 12

- **PPO Configuration**: Mini-batch size of 192, micro-batch size of 48
- **KL Penalty**: Coefficient $\beta = 0.001$ with low-variance KL loss formulation
- **Memory Optimization**: FSDP with parameter, gradient, and optimizer offloading enabled

## B.5 SEARCH SIMULATION AND RETRIEVAL

### B.5.1 TRAINING-TIME SEARCH SIMULATION

Following ZeroSearch Sun et al. (2025a), we employ a 14B parameter simulation LLM to generate search results during training, eliminating dependency on external APIs:

- **Simulation Model**: A fine-tuned LLM (`Simulation_LLM_google_14B_V2`) deployed via vLLM Kwon et al. (2023)
- **Deployment**: Tensor parallelism across 2 GPUs with 90% GPU memory utilization
- **Throughput**: Maximum 1,024 sequences with optimized batching
- **Document Generation**: Controlled quality through prompt engineering with adjustable noise injection

### B.5.2 TEST-TIME REAL SEARCH

During evaluation, we use real Google Search API via SerpAPI for authentic retrieval:

- **Search Engine**: Google Search with top-5 results retrieval
- **API Configuration**: Rate-limited queries to avoid throttling
- **Result Processing**: Extract and concatenate relevant snippets up to 2,048 tokens

## B.6 DATASET CONFIGURATION

We utilize the ZeroSearch dataset Sun et al. (2025a) organized as follows:

- **Training Data**: Questions from diverse QA benchmarks stored in Parquet format
- **Validation Data**: Held-out test split for monitoring training progress
- **Data Loading**: Shuffled training dataloader with drop_last=True for consistent batch sizes
- **Prompt Processing**: Maximum prompt length of 4,096 tokens with truncation at word boundaries

## B.7 EVALUATION PROTOCOL

- **Validation Frequency**: Every 600 training steps
- **Checkpoint Saving**: Every 50 steps with best model selection based on validation performance
- **Evaluation Metrics**: Accuracy, average search counts (ST), and search standard deviation (SD)
- **Reward Function**: F1-score based verification for answer correctness

## B.8 TRAINING INFRASTRUCTURE AND COMPUTATIONAL COST

- **Hardware Configuration**: All experiments were conducted on a single server node equipped with eight NVIDIA A100 40GB GPUs. The workload was distributed as follows:
  - **Simulation Environment**: 2 GPUs were dedicated to running the vLLM-based search simulator.
  - **Model Training**: 6 GPUs were used for the main training loop.
- **Memory Utilization**:
  - The two simulation GPUs each operated at approximately 90% memory capacity.

- The six training GPUs each maintained an average memory utilization of approximately 90% throughout the training process.
- **Training Duration**: The full two-stage training required approximately 8 hours to complete.
  - **Stage 1 (Warmup)**: ˜4 hours (150 steps).
  - **Stage 2 (Annealing)**: ˜4 hours (100-150 steps).
- **Total Computational Cost**: The total compute for a complete training run is estimated at **64 GPU-hours**, derived from (6 training GPUs $\times$ 8 hours) + (2 simulation GPUs $\times$ 8 hours).

## B.9 REPRODUCIBILITY

To ensure reproducibility of our results:

- **Random Seeds**: Fixed random seed (42) for model initialization
- **Code Release**: Full training and evaluation code will be made available upon publication
- **Model Checkpoints**: Trained model weights for both Stage 1 and Stage 2 will be released
- **Logging**: Comprehensive tracking via Weights & Biases for all experiments
- **Environment**: Docker container with exact package versions provided

## B.10 KEY IMPLEMENTATION DIFFERENCES FROM BASELINES

Our implementation differs from existing approaches in several crucial aspects:

- **Reward Formulation**: Unlike ZeroSearch which uses uniform search rewards, we employ adaptive sigmoid-based rewards that dynamically adjust based on group performance
- **Curriculum Design**: Explicit two-stage training with different reward coefficients and noise levels, rather than continuous annealing
- **Search Variance Regularization**: Novel component to promote behavioral consistency across identical queries
- **Thinking Action Counting**: Count discrete thinking actions rather than total length to preserve response diversity

# C EXPERIMENT SETUP

## C.1 BENCHMARKS

We evaluate our framework on a diverse set of question answering benchmarks to assess its search and reasoning capabilities across varying complexity. The benchmarks are categorized as follows:

- **Single-Hop Question Answering**: These benchmarks require retrieving a single piece of information to answer the question. We use:
  - **Natural Questions (NQ)** Kwiatkowski et al. (2019): Questions posed by real users to Google search.
  - **TriviaQA** Joshi et al. (2017): A challenging dataset of trivia questions.
  - **PopQA** Mallen et al. (2022): A dataset of popular questions about entities.
- **Multi-Hop Question Answering**: These benchmarks require finding and reasoning over multiple pieces of information to construct the answer. We use:
  - **HotpotQA** Yang et al. (2018): A standard benchmark for multi-hop reasoning.
  - **2WikiMultiHopQA** Ho et al. (2020): A more complex multi-hop dataset derived from Wikipedia.
  - **Musique** Trivedi et al. (2022): A dataset focusing on questions that require reasoning over multiple paragraphs.
  - **Bamboogle** Press et al. (2022): A dataset of challenging questions designed to be difficult for standard search engines.

For all benchmarks, we follow standard practice and use Exact Match (EM) as the primary evaluation metric.

## C.2    BASELINES

We compare Light-Search against a comprehensive set of baselines to evaluate its effectiveness and efficiency.

- **Direct Answer**: This is a zero-shot baseline where the model is prompted to answer the question directly without any explicit reasoning steps or external information. It measures the model's inherent knowledge.

- **Chain-of-Thought (CoT)** Wei et al. (2022): We prompt the model to generate a step-by-step reasoning process before providing the final answer. This baseline tests the model's reasoning capabilities without external retrieval.

- **Standard RAG** Lewis et al. (2020): A standard retrieval-augmented generation setup. For each question, we perform a one-time retrieval using a search engine and provide the retrieved documents as context to the model for answer generation.

- **ZeroSearch** Sun et al. (2025a): A state-of-the-art RL-based framework for training RAG models. It introduces a search simulator to avoid expensive real-time API calls during training and uses a curriculum learning approach. Unlike our proposed Light-Search, its reward mechanism does not explicitly optimize for search efficiency. This serves as our primary RL baseline.

