# OpenReview forum: "Light-Search: Reducing Retrieval Cost in RAG via Curriculum-Based Policy Optimization"
_ICLR.cc/2026/Conference — ICLR 2026 Conference Withdrawn Submission_

### Official Review · Reviewer_Dj1z · 2025-10-31

**Soundness:** 2
**Presentation:** 2
**Contribution:** 2
**Rating:** 4
**Confidence:** 4

**Summary:**

This article designs a two-stage training process: the first stage encourages the model to explore and reason, while the second stage optimizes the model's retrieval efficiency and response performance. Compared with Zero-Search and the basic RAG method, it can improve the model's retrieval efficiency and response performance.

**Strengths:**

1. The results seem promising and outperform several baselines.
2. The training process also appears relatively stable, enabling the model to learn more optimal strategies.

**Weaknesses:**

1. There is a lack of comparison with more advanced baselines, such as Search-R1 and its subsequent related works.
2. There is insufficient comparison with more adaptive retrieval methods, and the introduction of related works is also inadequate.
3. The article is difficult to understand, but it essentially focuses on two-stage reward parameter tuning. This approach does not seem novel, and the idea of balancing efficiency and quality was not first proposed in this article.

**Questions:**

1. Why can your two-stage training solve the problem that the model tends to learn extreme retrieval strategies?
2. The overall introduction of the reward is quite confusing.

---

### Official Review · Reviewer_L1zo · 2025-10-31

**Soundness:** 2
**Presentation:** 3
**Contribution:** 2
**Rating:** 4
**Confidence:** 3

**Summary:**

This paper proposes Light-Search, a reinforcement learning framework designed to train cost-efficient agents for Retrieval-Augmented Generation (RAG). The authors introduce a two-stage curriculum learning strategy to overcome the instability of directly penalizing search actions. The method first builds search capabilities in a cost-agnostic phase, then refines the policy for efficiency using a novel performance-gated reward function. Empirical results on question-answering tasks demonstrate a significant reduction in retrieval costs while maintaining competitive performance.

**Strengths:**

- Well-Motivated Problem and Approach: The paper effectively frames a critical, practical issue with RAG systems in Section 1 and Figure 1. The two-stage curriculum (Section 3.1) is a principled and intuitive solution to the well-known policy collapse problem in cost-aware RL.
- Strong Empirical Evidence: The experiments are comprehensive, testing across different model families and task complexities (Tables 1 and 2). The results consistently show that Light-Search achieves a superior accuracy-cost trade-off compared to the ZeroSearch baseline.
- Convincing Ablation and Complementary Studies: The ablation study in Table 3 provides strong evidence that both curriculum stages are indispensable. Furthermore, the complementary study (Tables 4 and 5), which shows that the warm-up stage can improve the ZeroSearch baseline, is a key result that highlights the modularity and general utility of the proposed capability-building strategy.

**Weaknesses:**

- The methodology lacks a sensitivity analysis of several key components, making it difficult to assess the method's robustness. The paper does not investigate how the curriculum switch-point (the fixed number of iterations, M₁, mentioned in Section 3.1.2), the Stage 1 reasoning reward weight (α in Equation (3)), or the specific formulation of the performance-gating function (sigmoid(Aᵢ⁺) - 0.5 in Section 3.2) affect the final policy's stability and performance. Crucial ablation studies are missing for these design choices, which would be necessary to understand their individual contributions and justify their specific configurations.
- The cost function C(τ), which is central to the Stage 2 objective, is vaguely defined in Section 3.1.2 as "a function of S(τ) and T_len(τ)". The paper provides no details on the relative weighting or normalization of these two distinct cost sources, which is a critical omission for reproducibility.
- The performance-gated reward in Stage 2 (Section 3.1.2) links the value of search directly to the final answer's quality. This creates a potential risk: on difficult questions where failure is common, the model may be penalized for necessary searches, leading it to become risk-averse and under-explore. Conversely, on easy questions where success is likely, it may reward redundant searches, leading to over-searching.
- The paper primarily uses "search count" as a metric for inference cost. While this is a reasonable metric it does not capture the full picture of efficiency. It remains unclear whether the model compensates for fewer searches by increasing its own generation length or complexity. This could potentially offset the gains from reduced retrieval, especially in terms of total wall-clock latency. The current evaluation does not rule out this possibility.
- The paper contains several minor formatting inconsistencies. For example, there are some issues with quotation mark styles ("retrieve-then-read" in Section 2.1) and inconsistent citation formats (e.g., GRPO () in Section 3.4 is an incomplete reference).

**Questions:**

See the weaknesses section.

---

### Official Review · Reviewer_godP · 2025-10-31

**Soundness:** 2
**Presentation:** 3
**Contribution:** 3
**Rating:** 4
**Confidence:** 3

**Summary:**

This paper presents Light-Search, a curriculum-based reinforcement learning framework that trains cost-aware retrieval-augmented generation (RAG) agents. The authors aim to make retrieval efficient rather than purely accurate. They design a two-stage learning curriculum called Two-Stage Advantage Shaping (TSAS). In the first stage, the capability-building phase, the model learns how to use retrieval effectively without any cost penalty; its advantage function includes group-normalized task rewards and intrinsic rewards for search depth and reasoning length. In the second stage, the efficiency-cultivating (annealing) phase, a performance-gated reward introduces cost awareness only for successful trajectories, encouraging the agent to preserve useful searches while penalizing redundant retrievals. The entire curriculum is optimized with a policy-gradient method named Light-Search Group-Relative Policy Optimization (LS-GRPO)—a customized variant of the GRPO algorithm. LS-GRPO operates on group-relative advantages to stabilize updates and control variance. Experiments on seven QA benchmarks (including NQ, TriviaQA and HotpotQA) show that Light-Search achieves up to 76.5 % reduction in retrieval operations while maintaining or slightly improving answer accuracy. Ablation studies confirm the necessity of both curriculum stages and the GRPO-based optimizer, and transfer experiments demonstrate that the warm-up stage can benefit other RAG systems such as ZeroSearch.

**Strengths:**

- The paper defines a clear and relevant problem: reducing retrieval cost in RAG while preserving accuracy. The motivation is specific, technically grounded, and practically important.
-  The proposed two-stage curriculum provides a structured learning process—capability building followed by efficiency tuning—that effectively avoids instability seen in direct cost-penalized reinforcement learning
- The performance-gated reward design links retrieval efficiency with answer quality in a controllable way, enabling a gradual transition from exploration to optimization.
- The adaptation of GRPO into LS-GRPO is technically sound; group-relative normalization reduces gradient variance and stabilizes training, as shown in reported learning curves.
- Experiments are comprehensive, covering seven QA datasets and several strong baselines. Reported improvements in both accuracy and retrieval cost are consistent and statistically reliable.

**Weaknesses:**

- The paper introduces a modified reward with a performance-gating function but provides no formal discussion of its optimization properties. There is no analysis of whether the new objective preserves an unbiased policy-gradient estimator or guarantees monotonic improvement, which weakens the theoretical rigor of the method.
- While ablations remove the warm-up or annealing stages, the paper does not isolate the effect of the gating function or cost term. It remains unclear which component contributes most to the observed cost–efficiency trade-off.
- The evaluation covers only single-hop and multi-hop QA datasets. It is unclear whether the proposed curriculum and reward design would generalize to other retrieval-heavy tasks such as multi-turn dialogue.
- Although the model achieves a lower average number of retrievals, the paper does not analyze why certain queries trigger retrieval versus not. There is no visualization of search trajectories, difficulty-dependent behavior, or qualitative examples of retrieval decisions.

**Questions:**

see weakness

---

### Official Review · Reviewer_fhBX · 2025-10-31

**Soundness:** 1
**Presentation:** 3
**Contribution:** 1
**Rating:** 2
**Confidence:** 5

**Summary:**

This paper proposes Light-Search, an RL framework that trains LLMs to balance answer quality with retrieval cost in RAG systems. The method employs a two-stage curriculum: (1) capability-building that rewards search exploration, and (2) efficiency-cultivating that introduces cost penalties through a performance-gated reward function. Experiments claim to reduce retrieval calls by up to 76.5% while maintaining competitive accuracy.

**Strengths:**

- Addresses an important practical problem—reducing RAG inference costs—which is critical for real-world deployment.
- The two-stage curriculum design is conceptually reasonable and provides clear motivation for avoiding naive cost penalties.
- Clear writing and comprehensive experimental coverage across multiple models (Qwen, LLaMA) and seven QA benchmarks.

**Weaknesses:**

### 1. Metric validity (critical)

The paper exclusively uses Exact Match (EM) as the evaluation metric. However, **EM doesn't capture search quality or strategic reasoning**. A model fine-tuned with EM could outperform prompting-based methods by memorizing patterns rather than learning when to search. The performance tables are not trustworthy for evaluating search capability.

### 2. Unfair baseline setup

The comparison with non-iterative baselines like RAG is imbalanced. If Light-Search performs multiple searches (e.g., 0.86 average might mean some queries get 2-3 searches), then RAG should receive equivalent total documents (e.g., top-10) for fair comparison. Otherwise, gains may stem from having more context rather than better search strategy.

### 3. Missing critical baselines

The paper omits essential baselines:
- **RAG + SFT**: Fine-tuning on retrieved documents with ground-truth answers (see "Baselines with retrievals" in Self-RAG [1]). This supervised baseline is widely recognized and necessary to attribute gains to the RL method.
- **Simple heuristic policies**: No comparison with rule-based strategies (e.g., "search if confidence is low").
- **Active RAG**: Cited but not compared experimentally.

### 4. Limited novelty

The technical contributions are incremental:
- Two-stage curriculum is standard RL practice
- Performance gating (sigmoid-based reward modulation) is a straightforward modification
- Combination of GRPO with curriculum, while effective, lacks algorithmic innovation

### 5. Missing interpretability and failure analysis

- No qualitative analysis of *when* and *why* the model searches vs. generates
- No case studies or failure analysis showing undersearch/oversearch patterns

**References**

[1] Asai et al., Self-RAG: Learning to Retrieve, Generate, and Critique through Self-Reflection, ICLR 2024.

**Questions:**

How sensitive are results to M₁, α, and other curriculum parameters? The paper provides no sensitivity analysis, making reproducibility uncertain.

---

### Official Review · Reviewer_hWNN · 2025-11-02

**Soundness:** 3
**Presentation:** 2
**Contribution:** 1
**Rating:** 2
**Confidence:** 4

**Summary:**

The paper introduces Light-Search, a reinforcement learning framework aimed at optimizing the retrieval process in Retrieval-Augmented Generation (RAG) systems. The authors identify a critical flaw in existing RL-based RAG systems: their exclusive focus on maximizing answer accuracy, which inadvertently leads to inefficient search behaviors. Light-Search addresses this by balancing task performance and computational efficiency, reducing retrieval calls by up to 76.5% while maintaining competitive task accuracy.

**Strengths:**

1.  The paper tackles a highly relevant and practical bottleneck for the real-world deployment of RAG systems. The motivation, clearly illustrated in Figure 1.
2. The core idea of decoupling capability-building from efficiency-tuning is sensible.

**Weaknesses:**

1. While the Light-Search framework presents some insights, many of the optimizations it introduces (such as improving retrieval efficiency and reducing latency) have been extensively explored in prior works, notably in the context of SearchAgent-X (Demystifying and Enhancing the Efficiency of Large Language Model-Based Search Agents). The absence of a detailed comparison with these existing baselines weakens the paper’s contribution.
2. Several key components of the method are not well-justified, making them hard to disentangle or reproduce. For example, the rationale behind the specific design of the "soft performance gate" used in Stage 2, which adjusts the cost-reward balance, lacks a solid theoretical foundation or comparison with other gating mechanisms. Without more detailed justification, it is challenging to understand the robustness of these strategies in different real-world contexts.

**Questions:**

See Weaknesses.

---

### Note · Authors · 2026-01-05

I have read and agree with the venue's withdrawal policy on behalf of myself and my co-authors.